# Smooth Autonomous Patrolling for a Differential-Drive Mobile Robot in Dynamic Environments

**DOI:** 10.3390/s23177421

**Published:** 2023-08-25

**Authors:** Ana Šelek, Marija Seder, Ivan Petrović

**Affiliations:** Laboratory for Autonomous Systems and Mobile Robotics (LAMOR), Faculty of Electrical Engineering and Computing, University of Zagreb, 10000 Zagreb, Croatia; ana.selek@fer.hr (A.Š.); ivan.petrovic@fer.hr (I.P.)

**Keywords:** mobile robot, patrolling, motion planning, path smoothing

## Abstract

Today, mobile robots have a wide range of real-world applications where they can replace or assist humans in many tasks, such as search and rescue, surveillance, patrolling, inspection, environmental monitoring, etc. These tasks usually require a robot to navigate through a dynamic environment with smooth, efficient, and safe motion. In this paper, we propose an online smooth-motion-planning method that generates a smooth, collision-free patrolling trajectory based on clothoid curves. Moreover, the proposed method combines global and local planning methods, which are suitable for changing large environments and enabling efficient path replanning with an arbitrary robot orientation. We propose a method for planning a smoothed path based on the golden ratio wherein a robot’s orientation is aligned with a new path that avoids unknown obstacles. The simulation results show that the proposed algorithm reduces the patrolling execution time, path length, and deviation of the tracked trajectory from the patrolling route compared to the original patrolling method without smoothing. Furthermore, the proposed algorithm is suitable for real-time operation due to its computational simplicity, and its performance was validated through the results of an experiment employing a differential-drive mobile robot.

## 1. Introduction

The patrolling task carried out by a mobile robot essentially boils down to a user-specified sequence of goal points that the robot must successively visit in order to achieve environmental monitoring and supervision.

Such robots have a wide range of applications [1,2] and the potential to replace or assist human operators in lengthy or dangerous real-world scenarios, such as mine clearance [3], nuclear environment monitoring [4], public security [5,6], military purposes [7,8], etc. In order to achieve efficient patrolling, the planned path program must satisfy some requirements, the most important of which are minimizing the patrolling execution time, path length, and robot energy consumption and enabling the ability to quickly replan a path when the environment changes during the execution of the patrolling task.

For patrolling, classical graph-based algorithms can be used, e.g., Dijkstra [9,10] or A* algorithms [11]. However, these algorithms are not suitable for changing environments, so local obstacle avoidance is required, e.g., the dynamic window approach (DWA) [12,13] or the use of an artificial potential field (APF) [14]. The disadvantages of DWA and APF include limitations in complex environments, local optimum traps, and sensitivity to parameter tuning. The D* algorithm [15], on the other hand, is suitable for changing environments because it can quickly recompute new paths to avoid obstacles. Graph-based algorithms are often implemented on a rectangular grid, and the resulting path consists of straight line segments with headings corresponding to grid transitions (45 degrees). The two-way D* (TWD*) algorithm computes more natural paths that are not limited to grid intersections and whose path segments are straight lines with arbitrary headings [16]. If the robot employed encounters an obstacle, a new TWD* path around the obstacle is calculated. The direction of the replanned path may have a large deviation from the current orientation of the robot. To solve this problem, various orientation alignment methods have been developed, and some motion-planning problems have been transformed into control problems [17,18,19]. These tasks can be computationally expensive, especially in complex and dynamic environments, which makes them difficult to use in real-world scenarios. Moreover, they can rely on models of a robot’s dynamics or a map-based representation of an environment, and any inaccuracies in these models can lead to suboptimal or infeasible paths or collisions with obstacles. Many motion-planning methods do not have continuity of curvature and generate jerky or discontinuous trajectories that require a robot to stop at each goal and align its heading with the orientation of the next path segment. This prolongs execution time, is energy consuming, and precludes such methods’ use for many real-world vehicles due to their kinematic constraints. These shortcomings can be overcome using smoothing algorithms.

Some methods that produce a smoothed path use simple shapes such as circles, arcs, and lines [20,21]. The presence of discontinuities at the intersections of straight lines and circular arcs makes these methods infeasible for real applications. A similar case applies for a cubic Hermite spline, which only has continuity of tangency [22]. Other methods based on, e.g., spirals, splines [23], and Bézier functions [24], have difficulty controlling curvature. This problem can be avoided by employing cubic splines [25] or clothoids. A parametric expression of a clothoid’s coordinates contains Fresnel integrals [26], which are transcendental functions that cannot be solved analytically, making them difficult to use in real-time applications. However, the very effective and computationally least-demanding way of calculating clothoids, and also the most suitable solution for real-time applications, is the method presented in [27]. So-called basic clothoid coordinates are stored in a lookup table, and the coordinates of any other clothoids (i.e., a general clothoid) are easily determined based on the stored basic clothoid coordinates via rescaling, rotating, and translating.

In this paper, we propose an online smooth-motion-planning method for a differential-drive mobile robot. The smooth-motion-planning method generates a smooth patrolling path that enables a robot to move in optimal time while following a smooth path, taking into account the kinematic and dynamic constraints of the robot, such as its maximum velocity and acceleration. Moreover, the proposed method is suitable for real-time operation due to its computational simplicity and since it allows for path replanning in case a robot encounters unknown obstacles.

In our previous work [28], a control architecture for the autonomous execution of GPS-based waypoint and patrolling tasks by a Komodo vehicle was presented. The patrolling task was achieved as an independent sequence of goal-finding tasks instead of planning the entire patrolling trajectory. In the cited paper, to avoid the in-place rotation of the robot, a smooth algorithm based on clothoids is used, and the smoothed path deviates from the patrolling path only at the goal points and in small regions around them. Here, we redesign our previous control architecture for differential-drive mobile robots and develop a combined global and local planner. The global planner is implemented as a user interface for task assignment and supervision in the Quantum geographic information system (QGIS). The local planner for obstacle detection and avoidance is based on a TWD* algorithm, and it is compared with DWA, APF, and a quickly reactive navigation approach based on model predictive control (MPC) [29]. The communication between QGIS and the Robot Operating System (ROS) is allowed via the client–server framework. The orientation alignment problem is solved using a new approach based on the golden ratio. This approach is a very effective and computationally undemanding way of calculating new commands for a mobile robot.

The key results and contributions of this paper have been summarized as follows:•The development of a complete autonomous navigation system for differential-drive mobile robots;•The development of a combined global and local planner for fast path replanning in dynamic environments;•The realization of real-time traversable collision-free path planning based on clothoids;•The realization of orientation alignment using the golden ratio.

The rest of this paper is organized as follows. Section 2 describes a theoretical framework for robot motion planning. Section 3 presents the architecture of the proposed robot control system that enables fully autonomous navigation. A pipeline of smooth patrolling and the integration of several algorithms are described in Section 4. The simulation and experimental results are given in Section 5 and Section 6, respectively, followed by the limitations and recommendations for the improvement of the proposed method in Section 7. The final Section 8 presents the conclusions.

## 2. Theoretical Background

Mobile robot navigation is a process of determining a collision-free motion from within the initial configuration in accordance with goal configuration while avoiding obstacles [30]. There are four basic components of this process, as presented in Figure 1:Perception—Sensors, e.g., GPS, encoders, IMU, 3D laser, cameras, etc., are used for collecting information from the environment for different applications, e.g., mobile robot localization, the mapping of the environment, and object or human tracking;Localization—The determination of a robot’s position in a global coordinate frame;Path planning and smoothing—Finding a feasible, smooth, collision-free path from the starting point to the goal point;Motion control—While respecting the actuator-related limits of the robot, the robot’s motions are controlled to ensure that it follows the desired trajectory.

### 2.1. Environment Model and Search Graph

An occupancy grid map is created via the approximate cell decomposition of the environment [31]. The whole environment is divided into square cells of equal size $D_{cell}$, wherein each cell contains weighted occupancy information. Cells are represented as a set of *M* elements M={1,…,M} with corresponding Cartesian coordinates of centers $c_{i}\in R^{2}$. Each cell contains the value o(i)=1 if it is free or o(i)=∞ if it is occupied. A set of free cells can be represented as N=M∖O, where O is a set of obstacle cells.

The weighted search graph G(N,E,W) is created from the occupancy grid map, where each free cell in the grid map is represented as a node in the graph. Two nodes i,j∈N are neighbors if the distance between them is equal to the cell size, i.e., ${∥c_{i}-c_{j}∥}_{\infty }=D_{cell}$. E is a set of edges *e* between neighbor nodes, and W is a set of weights *w*. All edges in the graph are weights, as expressed below:(1)$$w\left(e\right):=∥c_{i}-c_{j}∥max\left\{o\left(i\right),o\left(j\right)\right\}.$$

**Definition** **1**(Distance of a path)**.** *The weight or distance of the path P in a weighted undirected search graph G is defined as the sum of the weights of the edges along the path:*
(2)$$d\left(P\right):=\sum_{e\in P}w\left(e\right)\;$$
*If two nodes are not connected, we define the distance between them as d(i,j)=∞.*

**Definition** **2**(Optimal path)**.** *Let* Π *be the set of all existing paths between the start and goal nodes in a graph G. The path P∈Π is optimal if and only if*
(3)$$\begin{array}{ccc} d^{*}\left(start,goal\right):= & \min_{P} & d(P)\\ \; & s.t. & P\in \Pi \end{array}\;$$
*We can say that P is the shortest path from the start node to the goal node if it is optimal.*

### 2.2. Path Planning

All path-planning algorithms can be divided into two categories: offline and online. Offline algorithms use fixed information, and the environment is assumed to be known in advance. They are used for global path planning [9,10,11]. The global path is optimal, but these methods are not reactive to unknown or dynamic obstacles, and full prior knowledge of an environment is often impossible. On the other hand, online algorithms use real-time sensor measurements and are suitable for local path planning in changing environments [12,13,14]. Many approaches to robot navigation are based on a combination of global and local path planners: for example, in the proposed paper, a patrolling path is assigned via QGIS, and the local path planner is based on the TWD* algorithm.

#### Path Continuity

A path consisting of a sequence of straight lines leading from the start to the goal involves changes in the direction of motion. This path does not have continuous first and second derivatives, i.e., it belongs to $C^{0}$ paths with only continuity of position and not to $C^{1}$ (continuity of tangency) or $C^{2}$ (continuity of curvature). If the robot follows the $C^{0}$ path, it would exhibit step changes in its orientation, and by following the $C^{1}$ path, it would exhibit step changes in its angular velocity. These changes can result in a tracking error or collisions with obstacles in the worst case. Therefore, the path must be smoothed and become a $C^{2}$ path with continuity of curvature [32].

### 2.3. Path Smoothing

The clothoid coordinates with respect to its arc length *s* are yielded via the following expressions:(4)$$x\left(s\right)=x_{0}+\int_{0}^{s}\cos(\theta_{0}+\kappa_{0}\xi +\frac{1}{2}c\xi^{2})d\xi ,$$
(5)$$y\left(s\right)=y_{0}+\int_{0}^{s}\sin(\theta_{0}+\kappa_{0}\xi +\frac{1}{2}c\xi^{2})d\xi ,$$
where $\left(x_{0},y_{0},\theta_{0}\right)$ is the robot’s initial position; $\kappa_{0}=\frac{\omega_{0}}{v_{0}}$ is the initial curvature at s≥0, which denotes the arc length; and $c=\frac{\alpha }{v_{0}^{2}}$ is the sharpness, which describes how much the curvature changes with the traveled distance. $v_{0}$ and $\omega_{0}$ are the initial vehicle linear and angular velocities, respectively, and α denotes angular acceleration.

The curvature κ of the clothoid is expressed as follows:(6)$$\kappa \left(s\right)=\kappa_{0}+cs,$$

The tangent angle $\theta_{t}$ of the clothoid is expressed as follows:(7)$$\theta_{t}\left(s\right)=\theta_{0}+\kappa_{0}s+\frac{1}{2}cs^{2}.$$

Parametric expressions (Equation 4) and (Equation 5) of a clothoid curve contains Fresnel integrals [26], which are transcendental functions that cannot be solved analytically, making them difficult to use in real-time applications. Fresnel integrals are a pair of non-negative functions expressed as follows:(8)$$C\left(s\right)=\int_{0}^{s}\cos\left(\frac{\pi \xi^{2}}{2}\right)d\xi ,S\left(s\right)=\int_{0}^{s}\sin\left(\frac{\pi \xi^{2}}{2}\right)d\xi ,$$
Fresnel integrals will converge to a single point in the limit as the parameter *s* approaches infinity. With the argument $\frac{\pi \xi^{2}}{2}$, they converge to $\frac{1}{2}$, and with the argument $\xi^{2}$, they converge to $\frac{1}{2}\sqrt{\frac{\pi }{2}}$.

#### Feasible Paths

The motion of a mobile robot is limited by two constraints:A holonomic constraint—The wheels of the robot must roll and cannot slip:
(9)$$-\dot{x}\left(t\right)\cdot \sin\theta \left(t\right)+\dot{y}\left(t\right)\cdot \cos\theta \left(t\right)=0,$$The minimum turning radius of the robot is lower-bounded by the value $r_{min}$, and curvature κ is upper-bounded by $\kappa_{max}=1/r_{min}$.

If the path is a continuous sequence of configurations, it has a continuous curvature profile and corresponds to $C^{2}$, allowing the path to be smooth and allowing the robot to follow it with smooth control actions. A smooth path of finite length can be represented by the initial configuration q(0), its length *l*, and the curvature profile $\kappa :\left[0,l\right]⟼\left[-\kappa_{max},\kappa_{max}\right]$.

**Definition** **3**(Feasible, smooth, collision-free path)**.** *A smooth path $P_{s}$ is feasible and collision-free if it links the initial configuration $q_{s}\left(x_{s},y_{s},\varphi_{s},\kappa_{s}\right)$ to the goal configuration $q_{g}\left(x_{g},y_{g},\varphi_{g},\kappa_{g}\right)$ with a continuous function such that the following is true:*
*q(s):[0,l]⟼C, where $q\left(0\right)=q_{s}$, $q\left(l\right)=q_{g}$, and $q\left(s\right)\in C_{free},\forall s\in \left[0,l\right]$, where the configuration space of the mobile robot is expressed as $C\subset R^{2}xS^{1}$; $C_{free}$ is free configuration space without obstacles; and l is the length of the clothoid at the goal configuration;**The curvature profile is a continuous function $\kappa :\left[0,l\right]⟼\left[-\kappa_{max},\kappa_{max}\right]$;**The smooth path $P_{s}$ is collision-free: $\forall i\in \left\{1,\ldots ,o\right\},\forall s\in \left[0,l\right],q\left(s\right)\cap O_{i}=\emptyset$ where O is a set of obstacles.*

### 2.4. Motion Control

Motion control can be divided into two modules: trajectory planning and trajectory tracking. These two modules require knowledge about robot dynamics and actuator limits. The control action for a mobile robot is calculated according to a smooth path.

#### 2.4.1. Trajectory Planning

Time-optimal path planning requires a robot to drive at high speed, as this is necessary to render the path smooth [33]. The problem to be solved is a minimum time problem where time is calculated via the integration of time differentials along the path:(10)$$t=\int_{start}^{goal}\frac{ds}{v_{r}\left(s\right)},$$
where $v_{r}\left(s\right)$ is a robot’s velocity as a function of arc length. A time-optimal system requires a robot to always remain within its acceleration limits. To ensure time-optimal robot motion along a smooth path, the velocity profile with the maximum allowable velocities should be calculated. The maximum allowable velocity of a robot at a turning point (TP) is determined by the radial acceleration limit $a_{radial}$. The robot must decelerate and accelerate before and after the TP as much as maximally tangentially allowed by the acceleration limit. In this way, the maximum velocity profile is determined for each TP, start, and goal. The highest allowable overall velocity profile is determined as the minimum of all the velocity profiles.

#### 2.4.2. Trajectory Tracking

In this paper, we consider a differential-drive mobile robot for which linear and angular velocities must be calculated. The kinematic model of the differential-drive mobile robot can be represented as follows:(11)$$\left[\begin{array}{c} \dot{x}\left(t\right)\\ \dot{y}\left(t\right)\\ \dot{\theta }\left(t\right) \end{array}\right]=\left[\begin{array}{cc} \cos\theta (t) & 0\\ \sin\theta (t) & 0\\ 0 & 1 \end{array}\right]\left[\begin{array}{c} v(t)\\ \omega (t) \end{array}\right],$$
where *t* is time, (x(t), y(t)) is position, θ(t) is orientation, v(t) is linear velocity, and ω(t) is the angular velocity of the mobile robot. The trajectory-tracking algorithm finds the corrected linear and angular velocities, which are then transformed into the velocities of the robot’s wheels.

## 3. Control System Architecture

The architecture of the proposed robot control system is presented in Figure 2. The system consists of the user’s Windows computer, which is integrated with ROS navigation software on the robot’s Linux computer via the ZeroMQ communication protocol. The user’s Windows computer has a QGIS user interface for task assignment and the communication module connecting QGIS with ROS. Robot’s Linux computer contains all the ROS software packages required for navigation. Gray blocks are standard ROS packages or known inputs, while blue blocks are modules that we developed. In the following, three main modules are briefly described: task assignment and supervision deployed on the user’s Windows computer, the communication module deployed on both computers, and the autonomous navigation system deployed in the robot.

### 3.1. Task Assignment and Supervision

QGIS is a free, open-source geographic information system application designed to capture, prepare, store, manipulate, analyze, manage, and present geo-referenced data. In our previous work [34], the user interface for task assignment and the supervision of the robot was implemented as a QGIS plugin within which a user can choose between waypoint and patrolling tasks. A waypoint is a goal point in a map containing geo-referenced data pertaining to a location that the robot needs to reach in the physical world. Patrolling is a sequence of goal points, i.e., waypoints. Besides task assignment, the user can supervise the robot’s current position via the Home position and View positions requests.

### 3.2. QGIS and ROS Communication

To achieve efficient, robust communication between the user’s Windows computer and the robot’s Linux computer, the ZeroMQ library is used. ZeroMQ is based on the TCP/IP protocol and provides pre-optimized sockets that enable the flow of communication as messaging patterns. The Publish (PUB)/Subscribe (SUB) classic messaging pattern is used. On the user’s Windows computer, two SUB sockets are used, namely, one for the robot’s current position and one for positions during motion, and one PUB socket is used for task assignment. On the robot’s Linux computer, two PUB sockets and one SUB socket are used, (see Figure 2).

### 3.3. Autonomous Navigation System

The velocity controller calculates odometric characteristics by transforming the measured velocities of wheels into transnational and rotational velocity in the local robot frame, and vice versa. The extended Kalman filter (EKF) fuses the IMU and GPS data with the odometric data and yields the robot’s position and orientation in the global map frame. Two-dimensional laser range data are extracted from Velodyne point cloud data in order to ensure obstacles can be avoided during the execution of the smooth motion.

The smooth-motion-planning method provides the optimal linear and angular velocities that obey the kinematic and dynamic constraints of the robot, and this method is described in detail in Section 4.

## 4. Smooth-Motion-Planning Scheme

The pipeline of the proposed smooth patrolling scheme is shown in Figure 3, and each module is described briefly below. Our smooth patrolling scheme consists of two main parts: (i) a global planner that aims to accomplish the mission assigned to the robot and (ii) a local planner that determines a collision-free path to avoid newly detected obstacles.

### 4.1. Global Planner

Global path planning requires complete and deterministic knowledge of an environment to compute a path from the robot’s current position to the goal. For global path planning, we used patrolling, where routes are assigned to the robot in QGIS by the user (Section 3). The user is allowed to set an arbitrary route for the robot by selecting goal points on a map containing geo-referenced data. This assigned path acts as a global path. If an obstacle is not detected in the vicinity of the robot, this global path is directly forwarded to the path-smoothing module. Otherwise, if the obstacle is detected in the vicinity of the robot, the local planner is activated, and a replanned path is determined as described in Section 4.3.

### 4.2. Obstacle Detection

The inputs for the obstacle detection module are the patrolling path (PP), the smooth patrolling path (SPP), the position derived from EKF estimation, and the 2D laser range data derived from the Velodyne point cloud data (see Figure 3). We developed the following procedure for obstacle detection and goal selection that enables reactive navigation between obstacles.

First, laser readings are transformed from the local robot frame into the global map frame (see Algorithm 1, lines 3 and 4). Then, the vicinity of the starting position is checked. If an obstacle is detected in the vicinity of the starting position, i.e., if the distance global from the robot to obstacles is less than the predefined $d_{o}$, which is the minimum allowed collision-free distance, and it is equal to double the diameter of the circumscribed circle around the robot’s footprint, the algorithm selects the goal behind the robot, so the robot needs to move backward first. This goal is determined on the prolonged path segment behind the robot with a distance of $d_{o}$ from the obstacle as shown in Algorithm 1, lines 6–8.

If the obstacle is not detected in the vicinity of the starting position, determine whether the smooth path is collision-free. At every point $p_{s}$, compute the smooth path distance to obstacles in the global map frame. If the distance is less than the minimum allowed, i.e., if it is less than $d_{min}$, which is the width of the robot’s footprint, or if the robot’s current position is less than $d_{o}$, that is, the robot’s distance from obstacles in the global map frame, then the robot is on a collision course with an obstacle on the smooth patrolling path (see Figure 4). Calculate the replanned path only if the obstacle is on the current path segment, i.e., between two consecutive goal points (see Algorithm 1, lines 10 and 13).

Determine the goal for the local planner, accounting for the current position $p_{c}$ of the robot. For each two consecutive goals $p_{i}$ and $p_{i+1}$ along the patrolling path, the segment distance sd between goals is computed in $L_{1}$ norm as $sd={∥p_{i+1}-p_{i}∥}_{1}$ (see Algorithm 1, lines 18–21). Then, the distances $sd_{1},sd_{2}$ between the robot’s current position $p_{c}$ and goals $p_{i}$ and $p_{i+1}$ become $sd_{1}={∥p_{c}-p_{i}∥}_{1}$ and $sd_{2}={∥p_{i+1}-p_{c}∥}_{1}$. The patrolling path segment contains the robot’s current position if $sd=sd_{1}+sd_{2}$. Then, check if $p_{i+1}$ is not occupied with an obstacle, and select $p_{i+1}$ as the goal for the local planner. Otherwise, select $p_{i+2}$ as the goal for the local planner as performed in Algorithm 1, lines 22–26.
**Algorithm 1** Obstacle detection  1:**Input:** laserRanges, $p_{c}$, PP, SPP  2:**Output:** nextGoal  3:**for** ∀laserRanges **do**  4:    transform laserRanges from the local robot frame to the global map frame  5:**end for**  6:**if** $global<d_{o}$ **then**  7:    $nextGoalX=x_{c}+global\cdot cos\left(\varphi_{c}\right)$  8:    $nextGoalY=y_{c}+global\cdot sin\left(\varphi_{c}\right)$  9:**end if**10:**for** $\forall p_{s}\in SPP$ **do**11:    **for** ∀laserRanges **do**12:        **if** ${∥SPP-global∥}_{1}<d_{min}$ and ${∥global-p_{c}∥}_{1}<d_{o}$ **then**13:           replanning=114:        **end if**15:    **end for**16:**end for**17:**if** replanning **then**18:    **for** $\forall p_{i}\in PP$ **do**19:        sd = ${∥p_{i+1}-p_{i}∥}_{1}$20:        $sd_{1}$ = ${∥p_{c}-p_{i}∥}_{1}$21:        $sd_{2}$ = ${∥p_{i+1}-p_{c}∥}_{1}$22:        **if** $\left(sd_{1}+sd_{2}\right)\in \left(sd-0.5,sd+0.5\right)$ **then**23:           **if** $sd_{2}<d_{o}$ **then**24:               nextGoal=PP[i+2]25:           **else**26:               nextGoal=PP[i+1]27:           **end if**28:        **end if**29:    **end for**30:**end if**

### 4.3. Local Planner

For local path planning, the TWD* algorithm is adapted, which considers goals from global path planning, i.e., patrolling. The TWD* algorithm creates and searches graphs in occupancy grid maps of the environment. This algorithm is based on two D* searches called in two ways: forward from the start, and backward from the goal. The overall cost of the optimal path f(n) from the start to the goal is calculated as follows:(12)f(n)=h(n)+g(n)
where $h\left(n\right)=d^{*}\left(start,n\right)$ denotes the cost from the start to the node *n* (forward search), and $g\left(n\right)=d^{*}\left(n,goal\right)$ denotes the cost from the node *n* to the goal (backward search). Nodes with the minimal value *f* determine a polygonal area of minimal path costs, and through this area, the shortest possible geometrical straight-line path can be found with continuous headings.

The TWD* path contains points that are on the same straight line; to obtain only turning points, i.e., points where a change in path direction occurs, the filtering of collinear points must be performed. At these points, changes in the direction of motion of the robot occur. This path does not have continuous first and second derivatives, i.e., it belongs to $C^{0}$ paths with only continuity of position and not to $C^{1}$ or $C^{2}$. Following the $C^{0}$ paths would be difficult or even impossible for a nonholonomic mobile robot and can result in a tracking error and collisions with obstacles in the worst case. To avoid this problem, sharp transitions between two segments on the TWD* path can be smoothed to enable continuous motion.

The TWD* algorithm recalculates the new path from the robot’s current position to the next goal on the patrolling route that has not yet been visited nor occupied with obstacles. It assumes only local knowledge of the environment and does not need a map. The path around the newly discovered obstacle may have a sudden change in direction, which is why the robot must stop and align itself with the direction of the path. To avoid rotation in place, an orientation alignment algorithm is (used described in Section 4.4).

### 4.4. Orientation Alignment

The inputs for this algorithm are the TWD* path and the current position ($x_{c},y_{c},\varphi_{c}$) of the robot. The direction of the TWD* path, i.e., the replanned path around the obstacle, may have a large deviation from the current robot orientation, which is solved using the orientation alignment based on the golden ratio. To orient a robot in a given direction, this algorithm calculates extra points, which depend on the divergence between the direction of the first segment of the TWD* path and the orientation of the robot.

Two quantities $d_{1},d_{2}$, where $d_{1}>d_{2}$, are within the golden ratio if their ratio is the same as the ratio of their sum and the larger quantity:(13)$$\varphi =\frac{d_{1}}{d_{2}}=\frac{d_{1}+d_{2}}{d_{1}}=1.618.$$

The orientation alignment procedure (see Algorithm 2) is as follows:1.Determine the angle β as the direction of the first segment on the replanned TWD* path.2.Compute the absolute angular difference γ between the robot’s current orientation and the angle β. There are two possible cases: (i) 0° < γ≤ 90° and (ii) 90° < γ≤ 180°.3.For case (i), two additional points, $z_{0},z_{1}$, must be calculated (see Figure 5a). First point $z_{0}$ is always in the direction of the robot’s orientation and is at a distance of $d_{new}$ from the robot’s current position. $d_{new}$ is equal to the diameter of the circumscribed circle around the robot’s footprint. The second point $z_{1}$ is on the first segment of the replanned TWD* path determined using the golden ratio rule (Equation 13). For case (ii), three additional points, $z_{0},z_{1},P_{0}$, must be calculated (see Figure 5b). The procedure for calculating the first $z_{0}$ and third $z_{1}$ points is the same as for points $z_{0},z_{1}$ in the previous case. An additional point $P_{1}$ is added in the direction of the robot’s orientation minus 90° to reduce this case to the previous case.

After orientation alignment has been determined and a replanned path has been calculated, other non-visited points from the patrolling route are added and together input into the smoothing algorithm.
**Algorithm 2** Orientation alignment  1:**Input:** TWDPath, $x_{c},y_{c},\varphi_{c}$  2:**Output:** $z_{0},z_{1},P_{0}$  3:$\beta =atan2(y_{T_{1}}-y_{c},x_{T_{1}}-x_{c})$  4:$\gamma =|\varphi_{c}-$β|  5:$x_{z_{0}}=x_{c}+d_{new}\cdot \cos\left(\varphi_{c}\right)$  6:$y_{z_{0}}=y_{c}+d_{new}\cdot \sin\left(\varphi_{c}\right)$  7:**if** 90° < γ≤ 180° **then**  8:    $\gamma =\varphi_{v}-90$°  9:    $x_{P_{0}}=x_{c}+d_{new}\cdot \cos\left(\gamma \right)$10:    $y_{P_{0}}=y_{c}+d_{new}\cdot \sin\left(\gamma \right)$11:**end if**12:**if** 0° < γ≤ 45° **then**13:    $d_{1+2}=d_{1}\cdot 1.618$14:**else if** 45° < γ≤ 90° **then**15:    $d_{1+2}=d_{1}\cdot 1.618^{2}$16:**end if**17:$x_{z_{1}}=x_{c}+d_{1+2}\cdot \cos\left(\beta \right)$18:$y_{z_{1}}=y_{c}+d_{1+2}\cdot \sin\left(\beta \right)$

### 4.5. Path Smoothing

Smoothing is performed at the turning points of the path in order to avoid sharp turns and allow for continuous motion of the robot without stopping at these points. The use of clothoids allows for the determination of the shortest path, taking the prescribed constraints of acceleration and linear and angular velocities into consideration.

A clothoid can be discretized to equal intervals of Δs, and points can be stored in a lookup table. The stored points in the lookup table are used for the calculation of other so-called general clothoids with any initial condition $x_{0},y_{0},\phi_{0},\kappa_{0}$ and any sharpness *c*. Gaps are introduced at the end of each interpolation interval, but dense sampling gaps are not notable in real applications when using a small sampling interval Δs. Interpolation error *e* at a sampling interval Δs decreases with clothoid length and converges to zero because a clothoid is a spiral that converges to a point, as presented in Section 2.3 and Figure 6b, where for Δs = 1, there is a large interpolation error that converges to zero over the clothoid’s length, and for Δs = 0.1, the interpolation error is barely noticeable. In Figure 6, a difference between the exact and interpolated clothoid is hardly visible.

The feasibility of the smooth path in the proposed paper is evaluated by considering clothoids with a typical constraint of the maximum orientation change of $\phi_{max}=\pi /2$ and arc length $s_{max}$ = 5 m. This smoothed TWD* path has linear changes in curvature and the property of $C^{2}$ continuity, which ensures continuous path curvature, allowing the robot to traverse the smoothed path without stopping.

### 4.6. Velocity Profile Optimization

Time-optimal path planning requires a robot to drive at high speed. With regard to driving at high speed along a smooth path, the velocity profile with maximum allowable velocities should be calculated in accordance with the kinematic and dynamic constraints of the robot. For that purpose, we use the velocity optimization algorithm described in detail in [33]. The main idea of this approach is to limit overall acceleration:(14)$$a=\sqrt{a_{\mathrm{radial}}^{2}+a_{\mathrm{tang}}^{2}},$$
where $a_{\mathrm{radial}}=v\times \omega =v^{2}\kappa$ is the radial and $a_{\mathrm{tang}}=\frac{dv}{dt}$ is the tangential acceleration of the robot.

The turning points (TPs) of the smooth patrolling path are local extreme points of the smooth path’s curvature, where the absolute value of the curvature is locally maximal, i.e., at these points, the turning radius reaches a local minimum; consequently, the velocity is locally minimal. The maximum allowable velocity of the robot at a TP is determined according to the radial acceleration limit $a_{\mathrm{radial}}$ and is defined as follows:(15)$$v_{TP}=\sqrt{\frac{a_{\mathrm{radial}}}{|\kappa_{TP}|}}.$$

The robot must decelerate before and accelerate after each TP considering the acceleration limit. Additionally, a constraint on the maximum allowable velocity $v_{max}$ of the robot should also be imposed. As soon as the velocity constraint is violated, the robot must stop accelerating and continue moving at the following velocity: $v\left(t\right)\leq v_{max}$.

The outputs of the velocity profile optimization algorithm are the optimal linear and angular velocities of the robot, which, together with the robot’s position and orientation, result in the 5D trajectory ${\left(x,y,\theta ,v,\omega \right)}^{T}$. The 5D trajectory is the input fed to the trajectory-tracking algorithm based on the Kanayama controller [35]. This controller ensures that the robot tracks the optimal 5D trajectory via the rejection of robot position disturbance caused by the localization module and velocity measurements from the wheel encoders.

### 4.7. Trajectory Tracking

The robot’s tracking control procedure finds the linear v(t) and angular ω(t) velocity that achieve the control objective. The Kanayama controller uses the reference configuration $p_{r}$$={\left(x_{r},y_{r},\theta_{r}\right)}^{T}$, taken from the 5D trajectory at the current time instant, and the measured current configuration $p_{c}={\left(x_{c},y_{c},\theta_{c}\right)}^{T}$. Then, it computes the error configuration $p_{e}$ as the difference between $p_{r}$ and $p_{c}$, which must converge to zero. The expression for error configuration is as follows:(16)$$p_{e}=\left[\begin{array}{c} x_{e}\\ y_{e}\\ \theta_{e} \end{array}\right]=\left[\begin{array}{ccc} \cos\theta_{c} & \sin\theta_{c} & 0\\ -\sin\theta_{c} & \cos\theta_{c} & 0\\ 0 & 0 & 1 \end{array}\right]\left(p_{r}-p_{c}\right).$$
Using the error configuration $p_{e}$, the reference linear and angular velocities $v_{r}$, and $\omega_{r}$ from the 5D trajectory at the current moment in time, the corrected linear velocity *v* and the corrected angular velocity ω are calculated as follows:(17)$$\left[\begin{array}{c} v\\ w \end{array}\right]=\left[\begin{array}{c} v_{r}\cos\theta_{e}+K_{x}x_{e}\\ \omega_{r}+v_{r}\left(K_{y}y_{e}+K_{\theta }\sin\theta_{e}\right) \end{array}\right],$$
where $K_{x}$, $K_{y}$, and $K_{\theta }$ are positive constant parameters of the controller. In [35], a Lyapunov function was employed to prove that a closed-loop system is stable for any value of $K_{x}$, $K_{y}$, and $K_{\theta }$. The outputs of the Kanayama controller are the corrected reference linear *v* and angular ω velocities, which are transformed into the robot’s wheel velocities.

## 5. Simulation Results

We tested the proposed smooth-motion-planning method and compared it with the previous point-to-point navigation scheme in the patrolling task (the original patrolling method was presented in [34]). The simulations were performed on a laptop equipped with a six-core processor, 16 GB of DDR4 RAM, and a NVIDIA Geforce GTX 1060. The algorithms were implemented and tested on a Robot Operating System (ROS). We devised two scenarios: (i) smooth motion planning without replanning and (ii) smooth motion planning with replanning.

The first scenario was developed to compare the proposed smooth patrolling method with the original patrolling method that uses MPC navigation that solves the optimization problem of each control time step in consideration of the state and control constraints along the future time horizon [29]. The local-minima navigation function is used in the MPC optimization problem, which imposes no limitations on an obstacle’s shape and can easily adapt to dynamic changes in the environment if it is computed via the D* discrete graph search or a similar algorithm. The main strengths of MPC are its low computational complexity and ability to generate near-optimal trajectories with guaranteed convergence. Additionally, the B-spline, Bézier curve, cubic spline, cubic Hermit spline, and Dubins curve are smoothing algorithms selected for comparison with the proposed smoothing algorithm based on clothoids. A B-spline curve is independent from the number of control points and has local control of the curve, i.e., any modification of the path segment will not affect the shape of the entire smooth path. A Bézier curve uses control points and Bernstein basis functions to define the shape of a smooth path. Cubic splines are piece-wise cubic functions, i.e., splines of minimum degree and sufficiently smooth in the presence of a small degree of curvature. A cubic Hermit spline requires end points and the slope at each end point to be specified. A Dubins curve produces the shortest curvature-constrained smooth path consisting of straight lines and circular arcs.

The second scenario was devised to demonstrate the fast recomputation of new smooth collision-free paths with the local planner and to show the effectiveness of obstacle avoidance. The DWA, APF, and MPC methods were selected for comparison to the proposed local planner based on the smooth TWD* algorithm since these approaches are reactive and efficient in dynamic environments. DWA is a local reactive avoidance method that takes into account the kinematic and dynamic constraints of a mobile robot and calculates the optimal trajectory at the current moment. APF is a path-planning method that allows a robot to navigate toward a goal while avoiding obstacles. The environment is represented as a virtual field of potential. The goal produces an attractive potential that pulls the robot towards the goal, while obstacles produce repulsive potentials that push the robot away. The total potential is calculated as the sum of the attractive and repulsive potentials. The robot’s movement toward the goal, which is determined using a gradient descent search method, serves as the negative gradient direction in the total potential field.

The driving limitations for velocity profile optimization were as follows: $v_{max}=0.5$ m/s and $\omega_{max}=0.5236$ rad/s are the maximum allowable linear and angular velocities, respectively, and $a_{\mathrm{radial}}=0.15$ m/s${}^{2}$ and $a_{\mathrm{tang}}=0.3$ m/s${}^{2}$ are the maximum allowable radial and tangential acceleration values, respectively.

The metrics chosen for the comparison of the path-planning algorithms are as follows:Path length (*L*)—calculated via the summation of the Euclidean distance between sampling points on the path;Execution time (*T*)—calculated via the summation of the discretization time required to travel between sampling points on the path;Tracking error ($e_{tr}$)—the deviation of the tracked trajectory from the patrolling route, which is determined as a surface between these two curves based on the trapezoidal rule;Average acceleration ($a_{avg}$)—calculated via the summation of acceleration values at each point on the trajectory;Average curvature ($\kappa_{avg}$)—calculated via the summation of all curvature values at each point on the trajectory;Initial path-planning time ($T_{init}$)—calculated via the average measured time required for ten algorithm executions.

In all the figures below, 2D laser data extracted from the Velodyne point cloud data are represented by green dots, the patrolling path selected in QGIS is shown as a blue line, the smoothed patrolling path is represented as a red line, and the tracked trajectory of the robot is denoted by a green line.

### 5.1. Smooth Motion Planning without Replanning

Figure 7a represents the patrolling route using the original patrolling method. At each goal point on the route, the robot stopped: its linear velocity was zero, and its angular velocity was maximal (see Figure 7c). The robot rotated in place until it aligned itself with the direction of the next patrolling segment.

Figure 7b represents the smoothed patrolling path. The deviation of the tracked trajectory from the calculated smooth path is negligible. The smooth path deviates from the assigned patrolling route only at turning points where two adjacent lines on the route are connected. The advantage of using the proposed approach can be seen in the velocity profile (see Figure 7d), where the robot’s linear velocity never drops below $v_{min}=0.125$ m/s and the robot’s tracked trajectory exhibits continuous motion without stopping. The curvature profile of the smooth patrolling path is represented in Figure 8, where the curvature changes linearly. The robot follows this path with curvature continuity without making abrupt and sharp turns.

The simulation results of the patrolling task with the original patrolling method and the proposed method are compared in Table 1. The proposed method yielded a reduced length of the tracked trajectory of around 3 m and its execution time was reduced by around 11 s compared to the results obtained with the original patrolling method. Moreover, the proposed approach presents a smaller tracking error compared to the original patrolling method. Lower average curvature means that the proposed method has a straighter smooth path, which results in smaller changes in angular velocity, and the path is followed in less time compared with the original method. Therefore, the average acceleration is also smaller when using the proposed method. The initial path planning with the original method is remarkable because the MPC navigation relay on the D* discrete graph search is computationally more demanding compared to the proposed method, which only smooths the patrolling path and calculates the velocity profile along this smooth path.

The clothoid-based path-smoothing procedure produces the $C^{2}$ path (see Figure 9a), and it exhibits continuity of curvature, which changes linearly over the path (see Figure 10a). The velocity profile is smooth, without exhibiting sudden changes in linear or angular velocity (Figure 11a). The B-spline- and Cubic-spline-based path-smoothing methods produce a $C^{2}$ path and present continuity of curvature (see Figure 10b,d). The B-spline path is presented in Figure 9b, and the cubic spline path is presented in Figure 9d. The velocity profile is smooth and does not show sudden changes in velocity profile (see Figure 11b,d). The Bézier curve, cubic Hermit spline, and Dubins curve produce the $C^{1}$ path (continuity of tangency); the curvature of this path is discontinuous, and by following this path, the robot exhibits step changes in its angular velocity (see Figure 11c,e,f).

The additional metrics chosen for the comparison of the smooth path algorithms are as follows:Bending energy (BE)—calculated via the summation of the squares of the curvature at each point of the trajectory:
(18)$$BE=\frac{1}{n}\sum_{s=1}^{n}\kappa^{2}\left(s\right)$$
where κ is the curvature at each point of the robot trajectory and *n* is the number of points in the trajectory [36].Curvature variation energy (CVE)—calculated via the summation of the squares of the change in the curvature in the trajectory, i.e., the sharpness at each point of the trajectory:
(19)$$CVE=\frac{1}{n}\sum_{s=1}^{n}\kappa^{\prime }{}^{2}\left(s\right)$$

A comparison of the state-of-the-art smoothing algorithms is presented in Table 2. The proposed method based on clothoids yielded the shortest path, which the robot followed in the shortest time. The worst results were obtained using the B-spline curve, where the path length and travel time were the largest. Moreover, the proposed approach has the smallest tracking error among all the methods tested. The largest tracking error was obtained for the Bézier and B-spline curves. The value of the average curvature of the clothoid is similar to that of the cubic spline. For straighter paths, the BE and CVE values are lower; this result is desirable since the energy requirement is increased according to the increase in the curvature of the trajectory. BE and CVE were also the smallest for these two curves. The worst curvature, BE, and CVE were obtained using a B-spline since it has a large deviation at the beginning of the path and a sharp turn with high curvature (which is presented as a big spike in curvature profile) (see Figure 10b). These characteristics result in a negative spike in angular velocity, and the linear velocity drops to 0.03 m/s (see Figure 11b), which is similar to the value of in-place rotation.

### 5.2. Smooth Motion Planning with Replanning

To demonstrate the effectiveness of obstacle avoidance using our replanning algorithm and reveal how it works, we designed a scenario where a patrolling route is given so that the global path intersects with an obstacle. This scenario is represented in Figure 12a. When the robot encounters an obstacle, the local planner determines the replanned path. To avoid rotation in place, an orientation alignment algorithm based on the golden ratio is executed, provoking the robot to first move forward, turn, and continue along the replanned path (see Figure 12b black line). This replanned path, together with the other unvisited patrolling goal points, is sent to the smoothing algorithm, which smooths the new path (red line).

In Figure 12c, the reference velocity profile (blue line) is shown in advance, while the actual velocity profile (red line) is shown up to the time of the possible collision with an obstacle. After replanning (Figure 12d), the reference linear velocity (blue line) starts from zero for safety purposes since the robot needs to slow down before continuing to drive along the replanned path, and the actual linear velocity (red line) drops to the $v_{min}$ value. For clarity, the time in the replanning scenario is reset to zero.

Additionally, a more complex scenario (see Figure 13) was executed, where the robot avoided two dynamic obstacles in an unknown environment. In Figure 13a, the robot marked with *robot0* executes path planning with the STWD*, APF, DWA, and MPC algorithms to reach *goal0*. *Robot1* and *robot2* represent dynamic obstacles that drive forward to goals denoted as *goal1* and *goal2*, respectively, at a constant velocity v=0.15 m/s. The travel trajectories of *robot1* and *robot2* are represented as cyan dashed lines, and the travel trajectories of *robot0* are represented as green, red, magenta, and blue lines for the STWD*, APF, DWA, and MPC algorithms. The robot’s positions are marked with the robot’s footprint as dashed lines. When *robot0* detects *robot1* on the path within the range $d_{o}=2.2$ m, the local planner based on the STWD* algorithm is executed, and the new path around the obstacle is calculated (Figure 13b). At the same timestamp when STWD* path replanning is executed, the position of *robot0* along with the travel trajectories using APF, DWA, and MPC are presented in Figure 13b. The calculation of the replanned path using the STWD* algorithm lasts less than 0.1 ms, which is fast enough for the real-time use of the proposed method; additionally, collision with *robot1* was avoided in the simulation.Afterward, *robot0* encounters the second dynamic obstacle, and a new replanned path around this obstacle is calculated (Figure 13c). Figure 13d presents the collision-free travel trajectories of *robot0* from the start to *goal0*. Figure 14 presents the velocity profiles of travel trajectories using the STWD*, APF, DWA, and MPC algorithms. The STWD* algorithm has a smoother velocity profile compared to the other methods.

The results of the comparison of the STWD*, APF, DWA, and MPC algorithms are given in Table 3. The proposed method has a shorter travel trajectory and a reduced execution time compared to the results obtained with other methods. APF has the longest travel trajectory, and the robot did not reach the goal but only entered the vicinity due to the goal-nonreachable-with-an-obstacle-nearby (GNRON) problem. Average curvature was the highest for MPC because the robot rotated in place in the starting position (see Figure 14 blue line). DWA had the lowest curvature along the path. MPC outperformed the other methods in terms of average acceleration because it found the optimal controls over a prediction horizon by minimizing the navigation function computed from the D* path. The other algorithms yielded similar average acceleration values. The initial path planning with the proposed method was high because the TWD* algorithm performs two D* discrete graph searches and calculates the shortest path within the area of minimal path costs, which are computationally more demanding compared to the other local path-planning methods. APF yielded the lowest initial path-planning values from the start to the goal.

## 6. Experimental Results

The experiments (the smooth autonomous patrolling experiments are shown in the accompanying video available here https://youtu.be/9F-SzTUmzjo (accessed on 18 August 2023)) were conducted on a system (illustrated in Figure 15) consisting of a Husky mobile robot and a base station. The Husky mobile robot is equipped with an Intel NUC7i7BNH, the SMART-V1G Novatel integrated L1 GPS + GLONASS receiver and antenna, Xsense IMU, and Velodyne HDL-32e LiDAR. The NUC runs Linux Ubuntu 16.04 and it consists of an i7 CPU and 2x8GB of DDR4 RAM. Velodyne HDL-32e was used for map building, and GPS was employed to provide a precise global position in an outdoor environment. IMU was used for orientation estimation for all three axes (yaw, pitch, and roll). The base station consists of one PC equipped with an i7 CPU and 16 GB of RAM. The PC ran on Windows 10 and was used for assignments and monitoring missions based on QGIS.

The experiments were performed in our lab to show how the proposed method works in a small room with static and dynamic obstacles. We designed two scenarios: (i) smooth motion planning without replanning and (ii) smooth motion planning with replanning. The first scenario was designed for the comparison of the proposed smooth patrolling method with the original patrolling method. The second scenario was designed to demonstrate the global/local planner and show the effectiveness of obstacle avoidance in static and dynamic environments.

In all the figures below, the real obstacles are extracted 2D laser range data from the Velodyne point cloud data and are presented as green dots, while the patrolling path is denoted by a blue line, the smooth patrolling path is denoted by a red line, and the tracked trajectory from the start to the goal is denoted by a green line. The TWD* algorithm was used for obstacle avoidance, and the replanned TWD* path is represented by a black line. The start, replanning, and goal positions of the robots are marked with the robot’s footprint as dashed lines.

### 6.1. Smooth Motion Planning without Replanning

The results of smooth patrolling in a static environment without obstacle avoidance are shown in Figure 16. The patrolling task executed using the original method based on the MPC navigation is presented in Figure 16a. The MPC navigation function was computed using D*, which considers only four orthogonal neighbor cells in a grid map of the environment. This resulted in a path with sharp turns (see Figure 16c; yellow lines). However, the travel trajectory does not follow this D${}^{*}$ path but finds the optimal controls over a prediction horizon by minimizing the navigation function computed with reference to the D${}^{*}$ path costs. For this reason, the deviation of the travel trajectory from the patrolling path was significant. The smooth path deviates from the assigned patrolling route only at turning points where two adjacent lines on the route are connected and at the beginning when the orientation of the robot was not aligned with the first segment of the route. The deviation of the tracked trajectory from the calculated smooth path was negligible. The successful execution of the patrolling task is represented in Figure 16d.

The linear velocity of the robot never drops below $v_{min}=0.125$ m/s, which can be seen in the velocity profile in Figure 16f. The robot tracked the trajectory with continuous motion without stopping. This is an advantage over the original method, where the robot stops at each goal on the patrolling route (see Figure 16e).

The experimental results of the patrolling task with the original and the proposed method are compared in Table 4. The proposed method has a reduced tracked trajectory length of around 1 m and its execution time was reduced by around 20 s compared to the results obtained with the original patrolling method. Moreover, the proposed approach has a smaller tracking error compared to the original patrolling method. Lower average curvature means that the proposed method has a straighter smooth path that can be followed in a shorter time compared with the original method. Therefore, the average acceleration is also smaller and exhibits fewer decelerations and accelerations at each goal on the patrolling path. The initial path planned with the proposed method is shorter compared to that of the original method because of the MPC navigation relay on the D* discrete graph search, which is computationally more demanding.

### 6.2. Smooth Motion Planning with Replanning

To demonstrate the effectiveness of obstacle avoidance in unknown environments, we designed two scenarios: (i) smooth motion planning with static obstacle avoidance and (ii) smooth motion planning with dynamic obstacle avoidance. The static obstacle was a box that was randomly placed somewhere in the environment, while another Husky mobile robot was used as a dynamic obstacle, which we controlled with a joystick.

The first scenario shows the effectiveness of the proposed method in a static environment where the patrolling route is assigned as the global path that collides with the static obstacle. This scenario is depicted in Figure 17. First, the patrolling route was assigned and the smooth patrolling path was calculated (Figure 17a). When the robot encounters a static obstacle, i.e., if the obstacle is detected on the smooth patrolling path within the range $d_{o}=2.2$ m, the local planner determines the replanned path around the obstacle, and collision is avoided (see Figure 17b). The replanned path is calculated with respect to the distance from the current position of the robot to the next unvisited goalon the patrolling route. This replanned path, together with the other unvisited goals from the patrolling route, is sent to the smoothing algorithm, which smooths the new path. The patrolling task ends when the robot arrives at the last goal on the patrolling route (see Figure 17c).

The velocity profiles for the *smooth motion planning with static obstacle avoidance* scenario are shown in Figure 18. When the obstacle is detected on the smoothed path, replanning is executed, and the robot slows down before continuing to drive along the replanned path. This is shown at around 15 sec, where the reference (blue line) and actual (red line) linear velocity drops to zero. Afterward, new linear and angular velocity profiles are calculated for the replanned path.

The second scenario shows the effectiveness of the proposed method in a dynamic environment. This scenario is represented in Figure 19. First, for the assigned collision-free patrolling route, the smoothing algorithm calculated a smooth patrolling path ( Figure 19a). When the robot encounters a dynamic obstacle, i.e., if the obstacle is detected on the smooth patrolling path within the range $d_{o}=2.2$ m, the local planner determines the replanned path around the obstacle, and collision with the obstacle is avoided (see Figure 19b). The replanned path is calculated with respect to the distance from the current position of the robot to the next unvisited goal on the patrolling route. For the replanned path, together with the last unvisited goal on the patrolling route, the smoothed path is calculated. The patrolling task ends when the robot arrives at the last goal on the patrolling route (see Figure 19c). The travel trajectory of the robot that represents the dynamic obstacle is shown as a magenta line.

The velocity profiles for the *smooth motion planning with dynamic obstacle avoidance* scenario are shown in Figure 20. When the dynamic obstacle is detected on the smoothed path, the replanning algorithm is executed, and the robot slows down before continuing to drive along the replanned path. This is shown at around 35 s, where the reference (blue line) and actual (red line) linear velocities drop to $v_{min}=0.125$ m/s. Afterward, new linear and angular velocity profiles are calculated for the replanned path. The results show that the robot can track the trajectory and successfully avoid collisions with continuous motion (i.e., without stopping).

Slight slippage occurs when a robot turns on a smooth curve on a flat and smooth surface, and this can be seen in the angular velocity profile as a deviation of the actual velocity from the reference velocity. This slippage is caused by various factors, such as the coefficient of friction between the robot’s wheels and the surface, the speed and change of the robot’s heading when driving on a smooth curve, etc.

## 7. Limitations and Recommendations for Improvement

In this section, we present a summary of the challenges and limitations related to the use of the proposed method in real-world applications. Furthermore, we propose solutions and future directions pertaining to the improvement of our method.

### 7.1. Hardware Limitation

First, the hardware limitation with respect to our experiment constituted the communication between the base station and the Husky mobile robot, which can be challenging in an outdoor environment if the WiFi connection breaks or its range is too short. This problem can be avoided by using better equipment like radio MPU5, which is frequently used in industrial applications.

### 7.2. Obstacle Detection

Software limitations may manifest if the robot drives on a natural, uneven, and rugged surface, e.g., snow, mud, forest ground, etc. Uneven terrain can create problems for the obstacle detection algorithm as we used only extracted 2D laser range data. A better solution may be using a mapping module, which can continuously build a 2.5D occupancy grid map of an environment based on Velodyne’s data. In a 2.5D occupancy grid map, local path planning based on smooth TWD* can provide better results.

### 7.3. The Velocity of Obstacle

The shortcoming of the proposed method is that we did not consider the velocity of dynamic obstacles, which we plan to account for in future work. We could use velocity information to directly determine potential collisions at some future time within a given time horizon. This would allow us to more precisely determine the side from which it is better to bypass dynamic obstacles and calculate the shortest replanned path with the TWD* algorithm. This can improve our method and reduce the risk of collision with dynamic obstacles.

### 7.4. Trade-Off between Path Planning Time and Accuracy

If the environment is large and the grid cells are small, the calculation of the replanned path can be extensive, because the time required for trajectory planning with the proposed method increases linearly with the number of nodes, while the number of nodes increases quadratically with the map dimensions. This can be avoided if the size of the grid cells is increased, but then the accuracy of the planned path may be reduced. So, for large environments, we could deal with a trade-off between the time required for the path replanning and the accuracy of the path. The possible improvement of the proposed method can be made in consideration of a global path or map of the environment if such a map is available. This can provide better path replanning if an obstacle is in the robot’s vicinity.

### 7.5. Different Mobile Robots

This method works for any type of nonholonomic or holonomic mobile robot, but small changes are required in accordance with the robot’s kinematic and dynamic constraints. For holonomic mobile robots, a trajectory-tracking controller can be modified such that it is efficient for a holonomic kinematic model, which is straightforward to determine.

## 8. Conclusions

The proposed patrolling algorithm is an online algorithm that generates a smooth, traversable, collision-free trajectory based on clothoids. It is computationally very efficient and suitable for path planning in real-time. Such a path is feasible for non-holonomic mobile robots since it does not contain sharp turns and it allows for the continuous motion of a robot without stopping at these points. By using a smooth technique on the patrolling path, efficiency can be significantly improved in terms of the required time and energy consumption during the patrolling tasks, as we showed in the comparison with the original patrolling method. Furthermore, the advantages of the proposed method are its robustness to environmental hazards and obstacles of different shapes and ability to quickly replan a new collision-free path if any change in the environment occurs. The simulation and experimental results show that our method can provide efficient and online navigation in static and dynamic environments. The simulation results demonstrate that the proposed local path-planning method yields the shortest travel trajectory and travel time from the start to the goal compared to the APF, DWA, and MPC algorithms. In the experimental results, a slight slippage occurred when the robot drove on a smooth curve due to a flat and smooth surface, which caused a slight deviation of actual angular velocity from the reference angular velocity.

For future work, more complex scenarios will be tested, where a robot will drive on a natural, uneven, and rugged surface, e.g., snow, mud, forested land, etc. We will consider how complex environments affect the proposed motion-planning method and work on its improvement. Furthermore, we will utilize tracks to prevent the wheels from slipping or skidding.

## Figures and Tables

**Figure 1 sensors-23-07421-f001:**
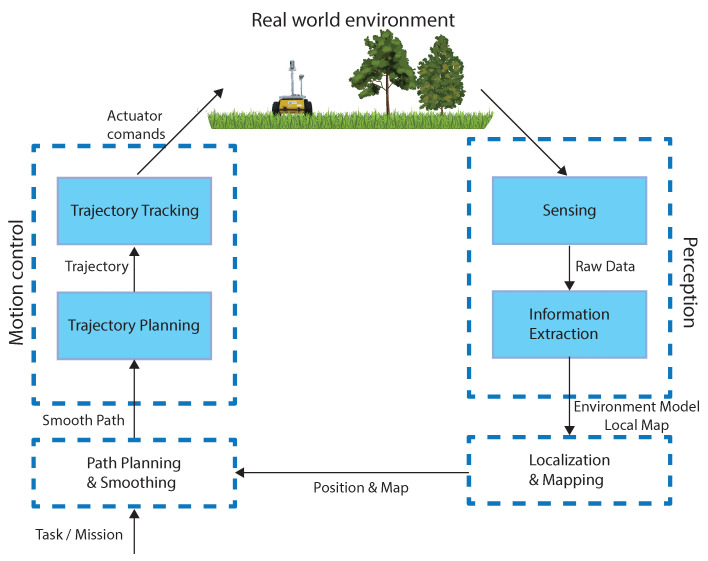
A theoretical framework for robot motion planning.

**Figure 2 sensors-23-07421-f002:**
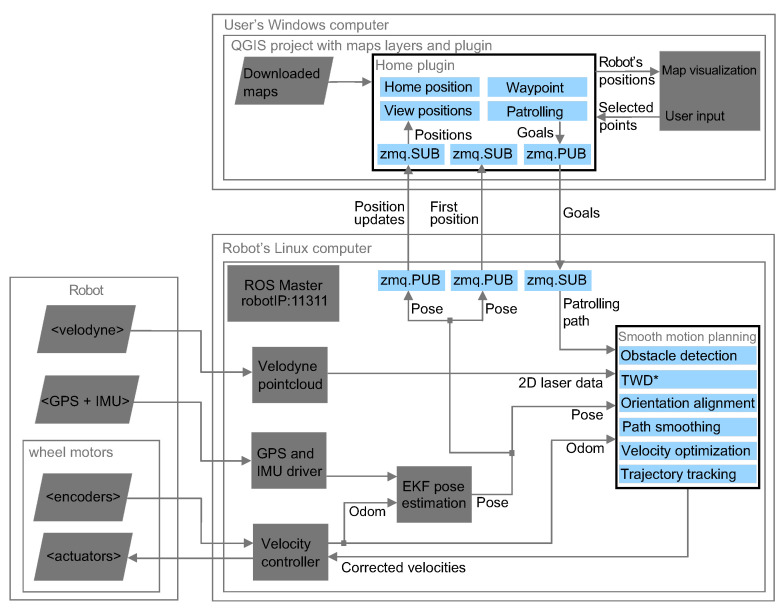
Control architecture of a differential-drive mobile robot.

**Figure 3 sensors-23-07421-f003:**
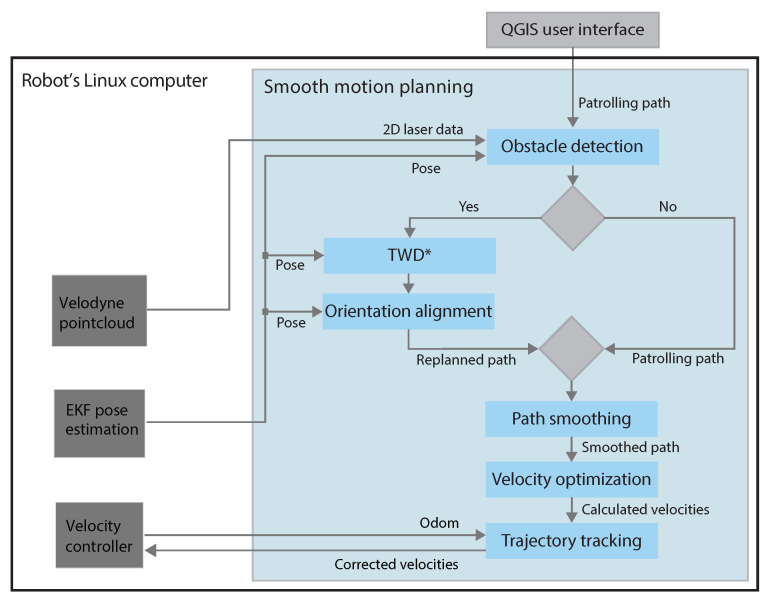
The pipeline of the smooth patrolling scheme.

**Figure 4 sensors-23-07421-f004:**
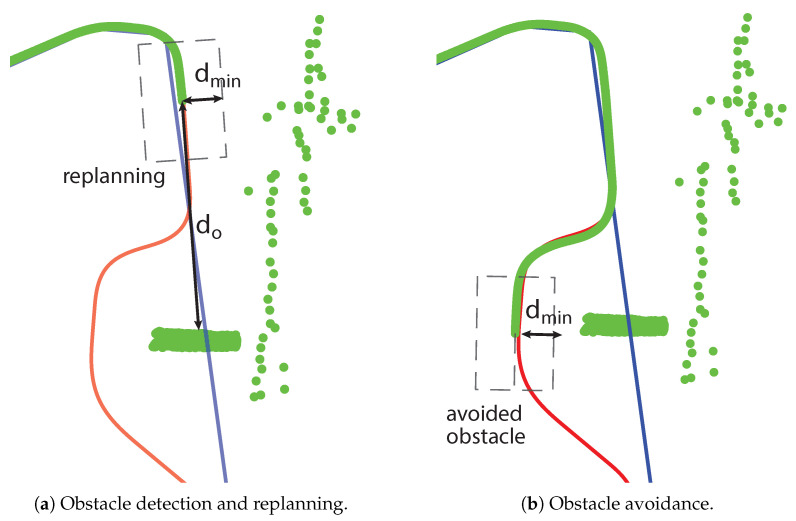
Two robots’ positions along with real obstacles (green dots), the patrolling route (blue line), the smoothed patrolling path (red line), and the tracked trajectory (green line).

**Figure 5 sensors-23-07421-f005:**
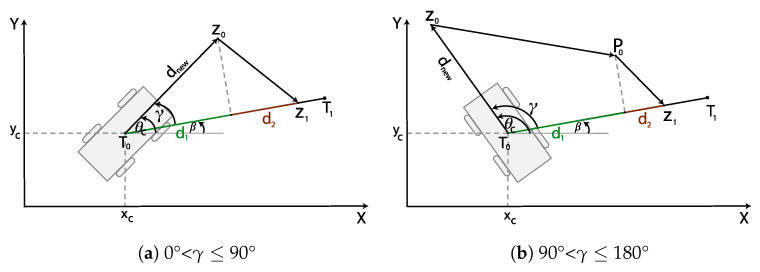
Orientation alignment is based on a golden ratio, where angle γ is the divergence between the robot’s current orientation $\varphi_{c}$ and the direction of the first segment β on the replanned TWD* path.

**Figure 6 sensors-23-07421-f006:**
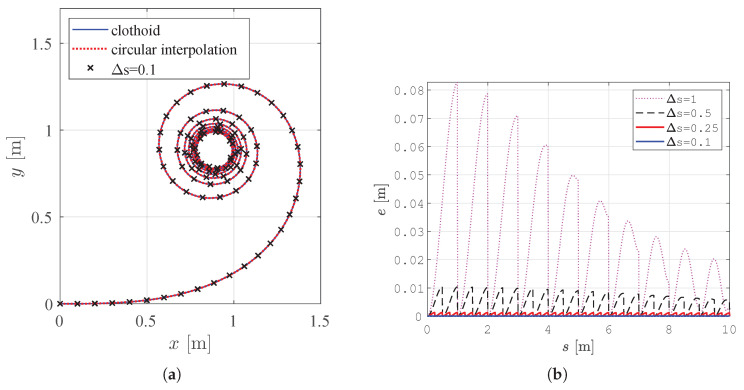
Clothoid with a length of 10 m interpolated via circular interpolation with the sampling interval Δs = 0.1 and interpolation error for different values of the sampling interval Δs. (**a**) Clothoid interpolated via circular interpolation. (**b**) Interpolation error *e* for different values of sampling interval Δs in the lookup table.

**Figure 7 sensors-23-07421-f007:**
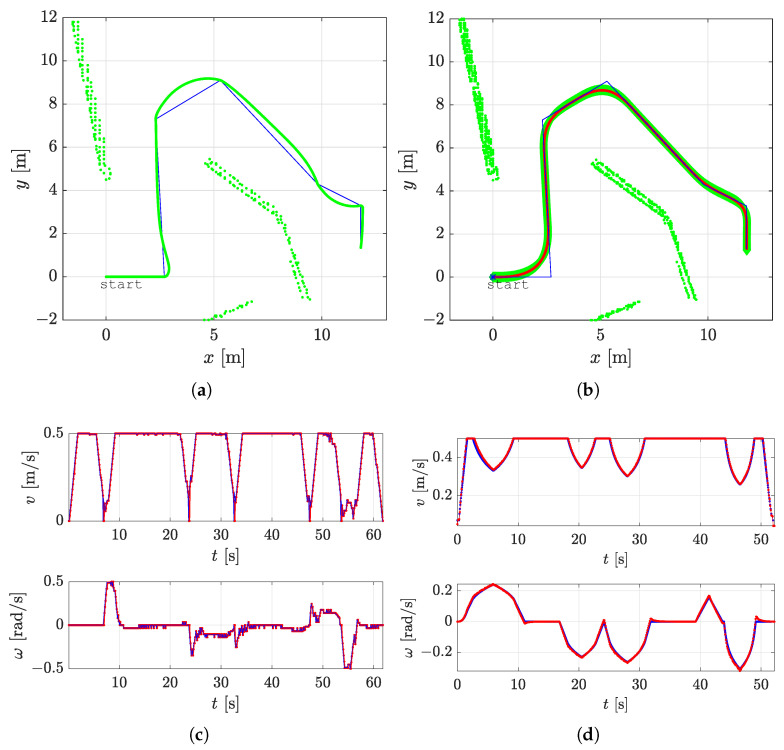
The comparison of the patrolling route (blue line) with the original patrolling method and the proposed smooth method, smoothed patrolling path (red line), the tracked trajectory (green line), the reference velocity profile (blue line), and the actual velocity profile (red line). (**a**) Patrol carried out using the original patrolling method. (**b**) Patrol carried out using the proposed smooth method. (**c**) Velocity profile yielded when using the original patrolling method. (**d**) Velocity profile yielded when using the proposed smooth method.

**Figure 8 sensors-23-07421-f008:**
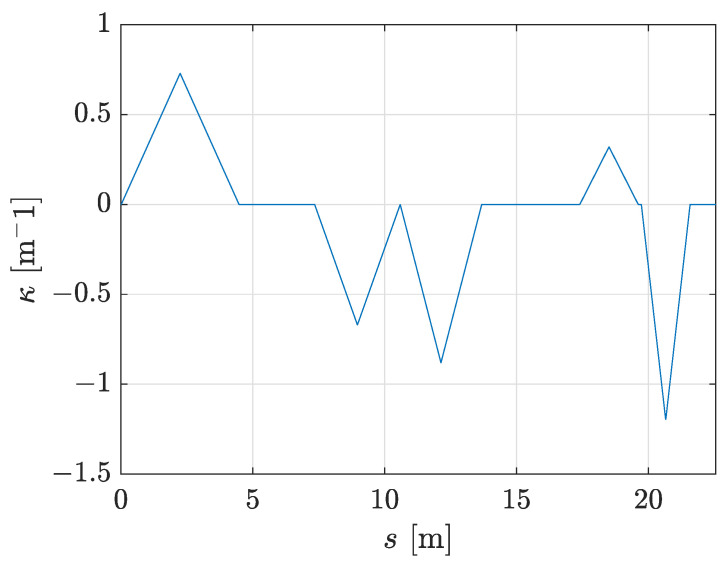
The curvature profile of the smooth patrolling path from Figure 7b.

**Figure 9 sensors-23-07421-f009:**
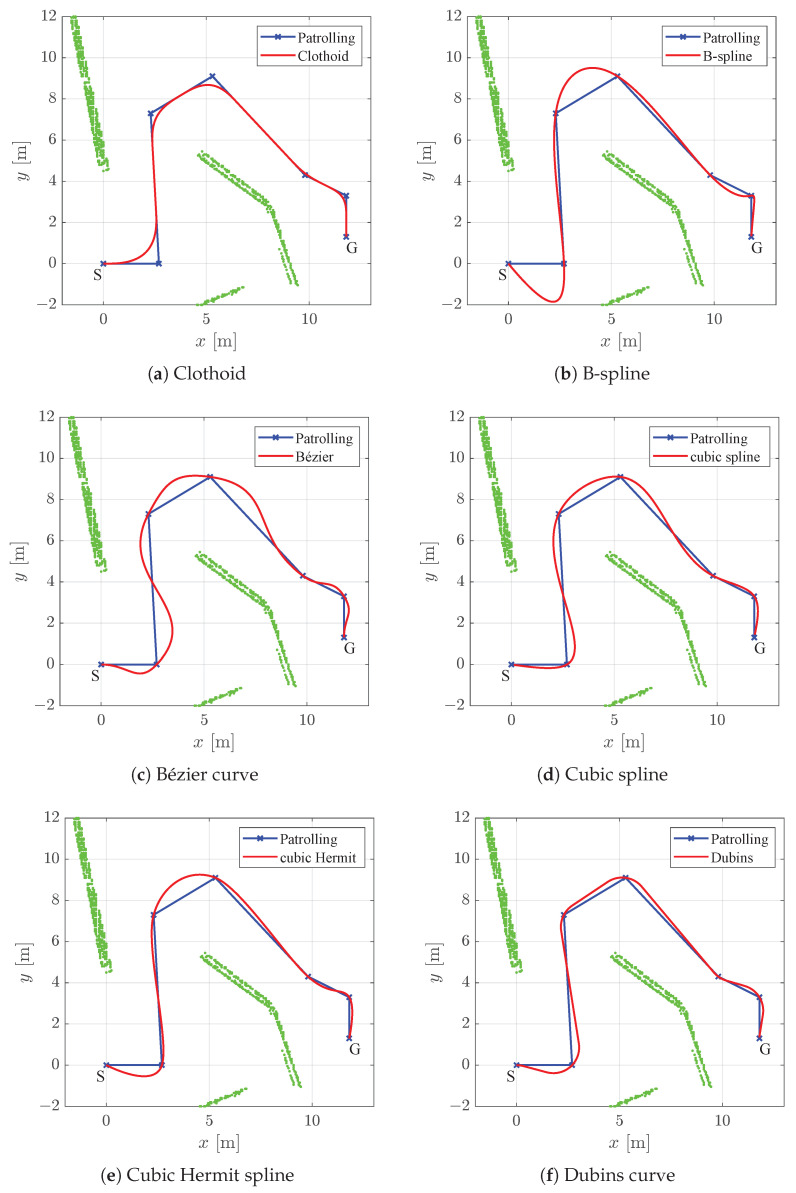
The smoothed path in a static environment for the proposed method and other state-of-the-art smoothing methods.

**Figure 10 sensors-23-07421-f010:**
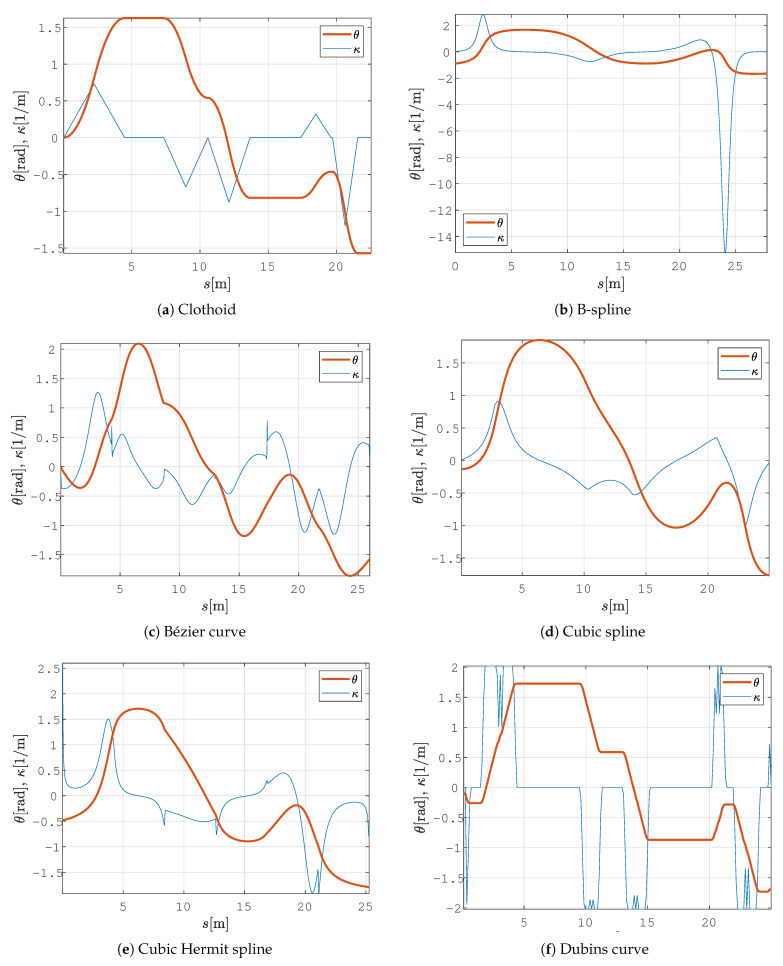
Curvature profiles and tangent angle change on a smooth path in a static environment with the proposed method and other state-of-the-art smoothing methods.

**Figure 11 sensors-23-07421-f011:**
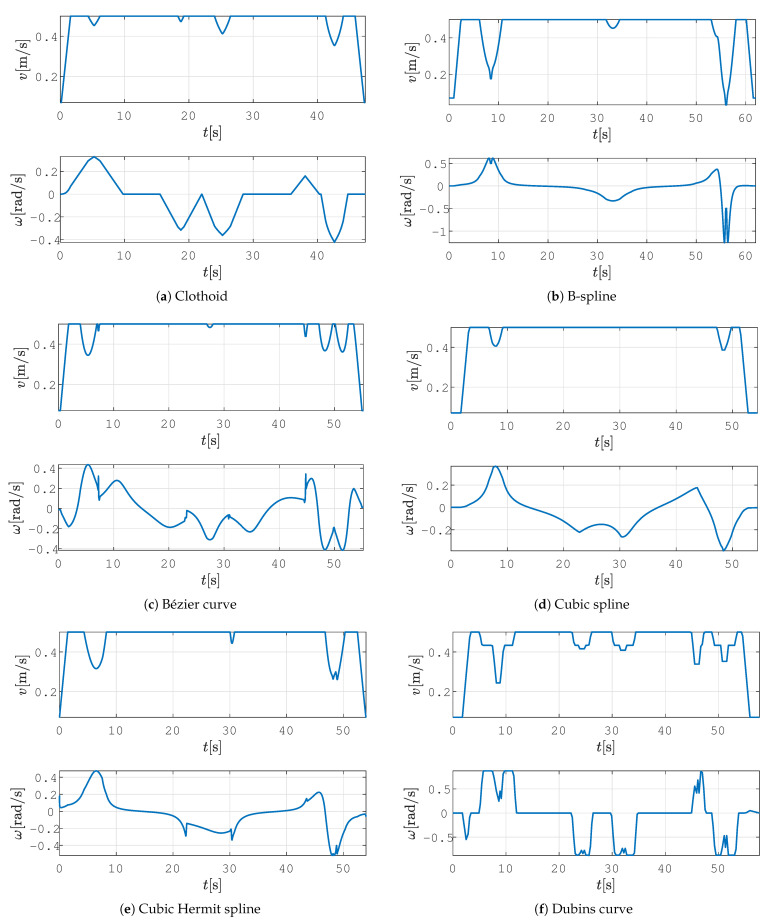
Velocity profiles on a smooth path in a static environment for the proposed method and other state-of-the-art smoothing methods.

**Figure 12 sensors-23-07421-f012:**
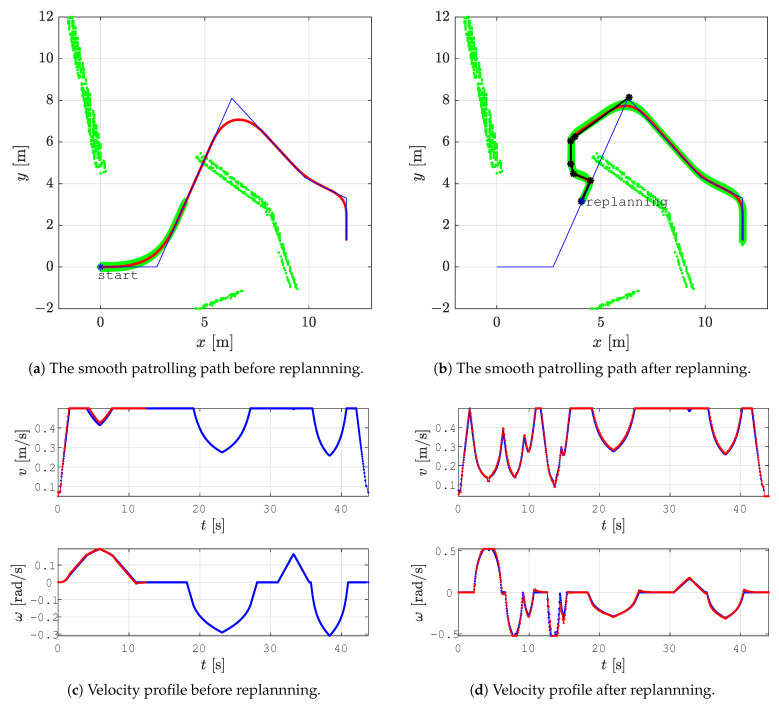
The patrolling route (blue line) during the execution of replanning, smoothed patrolling path (red line), the tracked trajectory (green line), the reference velocity profile (blue line), and the actual velocity profile (red line).

**Figure 13 sensors-23-07421-f013:**
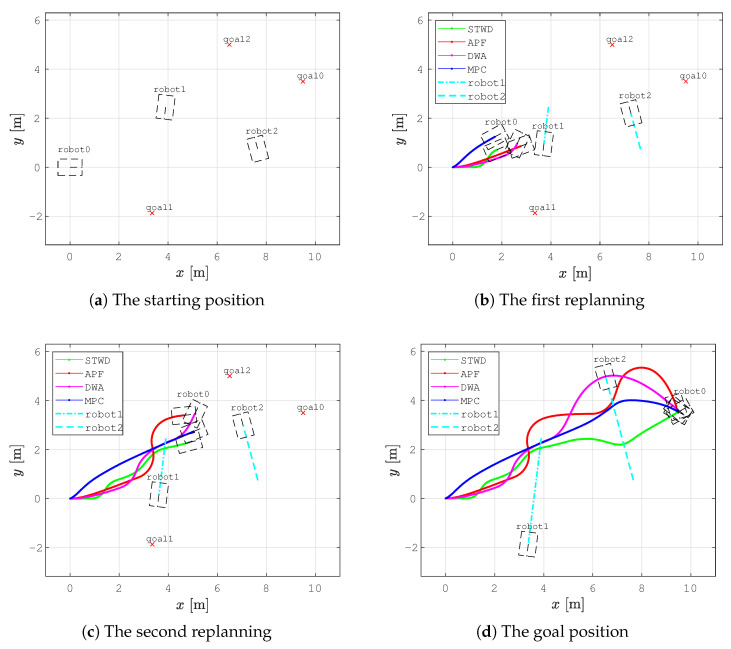
The step-by-step local path-planning scheme in a dynamic environment presenting the travel trajectories of *robot1* and *robot2* (cyan dashed lines) toward *goal1* and *goal2*, respectively: the STWD* (green line), APF (red line), DWA (magenta line), and MPC (blue line) travel trajectories of *robot0* from start to *goal0*.

**Figure 14 sensors-23-07421-f014:**
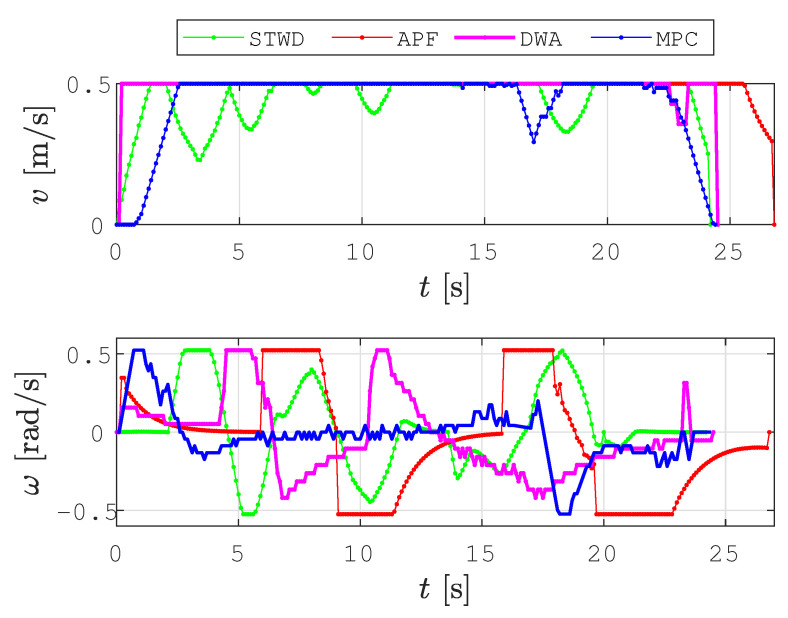
Velocity profiles of compared local path-planning algorithms executed in a dynamic environment: the STWD* (green line), APF (red line), DWA (magenta line), and MPC algorithms (blue line).

**Figure 15 sensors-23-07421-f015:**
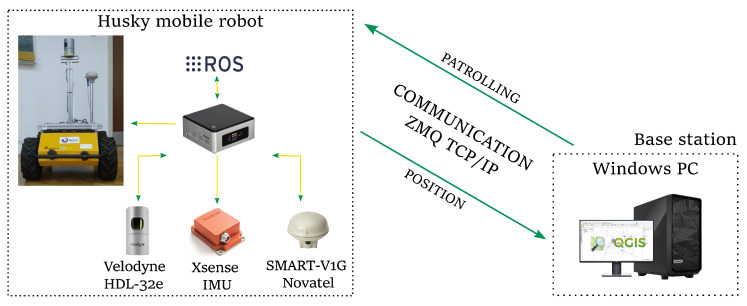
Experimental setup for smooth autonomous patrolling using the Husky mobile robot.

**Figure 16 sensors-23-07421-f016:**
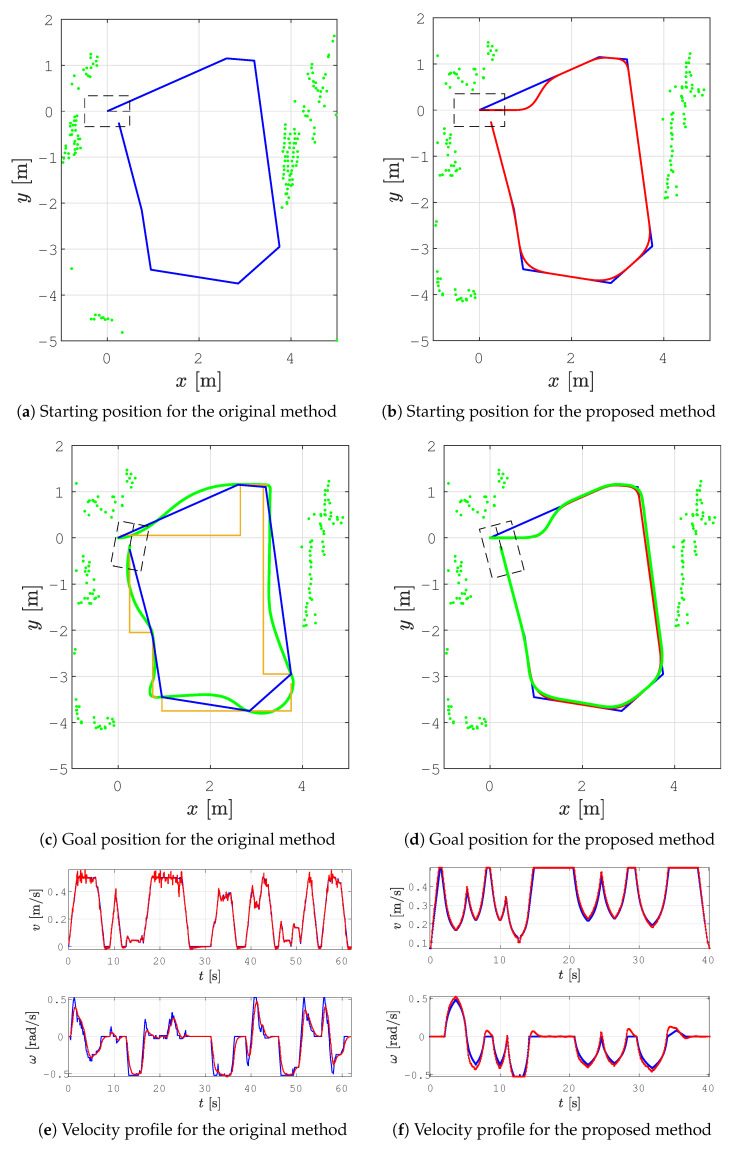
A comparison of patrolling route (blue line) traveled using the original and the proposed smooth methods: D* path (yellow line), smoothed patrolling path (red line), the tracked trajectory (green line), the reference velocity profile (blue line), and the actual velocity profile (red line).

**Figure 17 sensors-23-07421-f017:**
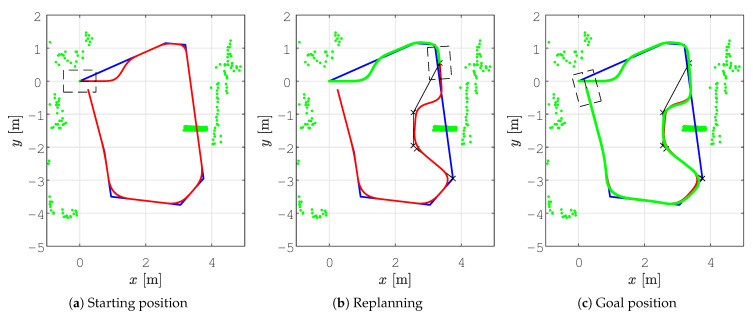
The step-by-step patrolling task executed in a static environment where the real obstacles are presented as extracted 2D laser range data from the Velodyne point cloud data (green dots), the patrolling path (blue line), the smooth patrolling path (red line), replanned TWD* path (black line), and tracked trajectory from start to goal (green line).

**Figure 18 sensors-23-07421-f018:**
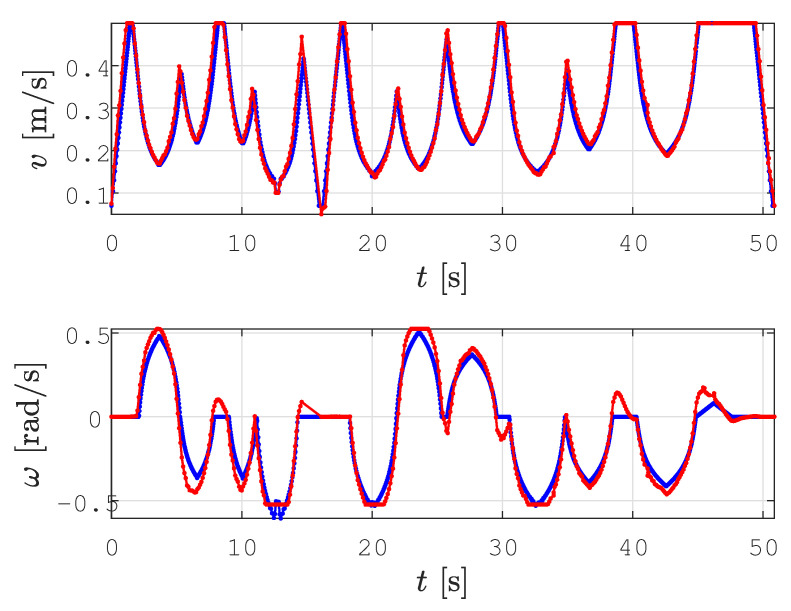
Velocity profile for the proposed method during the execution of the patrolling task in a static environment, presenting the reference velocity profile (blue line) and the actual velocity profile (red line).

**Figure 19 sensors-23-07421-f019:**
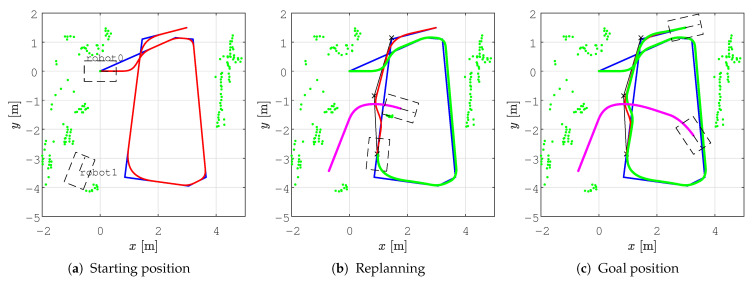
The execution of the step-by-step patrolling task in a dynamic environment, where the real obstacles are presented as extracted 2D laser range data from the Velodyne point cloud data (green dots), along with the patrolling path (blue line), the smooth patrolling path (red line), replanned TWD* path (black line), tracked trajectory from start to goal (green line), and the dynamic obstacle trajectory (magenta line).

**Figure 20 sensors-23-07421-f020:**
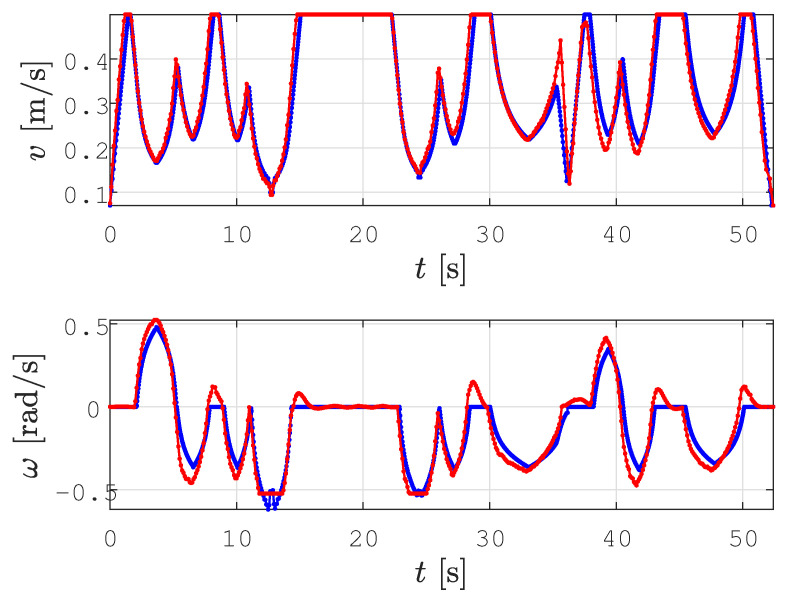
Velocity profile for the proposed method during the execution of the patrolling task in a dynamic environment, presenting the reference velocity profile (blue line) and the actual velocity profile (red line).

**Table 1 sensors-23-07421-t001:** Table comparing paths followed with and without smooth algorithm.

	Original Method	Proposed Method
L [m]	25.21	22.55
T [s]	61.8	51.6
$e_{tr}$ [m${}^{2}$]	4.32	3.67
$a_{avg}$ [m/s${}^{2}$]	0.24	0.08
$\kappa_{avg}$ [m${}^{-1}$]	0.94	0.24
$T_{init}$ [ms]	407.56	6.27

**Table 2 sensors-23-07421-t002:** Smoothing algorithm comparison table.

Smoothing Method	*L* [m]	*T* [s]	$e_{\mathrm{tr}}$ [m${}^{2}$]	$\kappa_{\mathrm{avg}}$ [m${}^{-1}$]	BE	CVE
Clothoid	**22.55**	**47.16**	**3.48**	**0.25**	0.14	0.25
B-spline	27.79	60.59	6.05	0.84	5.55	20.21
Bézier curve	25.97	54.66	8.25	0.44	0.28	1.67
Cubic spline	24.93	51.87	3.82	0.29	**0.13**	**0.11**
Cubic Hermit spline	25.28	53.96	3.48	0.40	0.35	4.16
Dubins curve	24.98	54.18	4.28	0.68	0.24	2.88

**Table 3 sensors-23-07421-t003:** Comparison table of local path-planning algorithms.

Method	L [m]	T [s]	$a_{\mathrm{avg}}$ [m/s${}^{2}$]	$\kappa_{\mathrm{avg}}$ [m${}^{-1}$]	$T_{\mathrm{init}}$ [ms]
STWD*	**10.70**	**24.20**	0.13	0.47	369.15
APF	13.17	26.72	0.13	0.55	**24.58**
DWA	12.07	24.45	0.12	**0.39**	49.45
MPC	10.93	24.40	**0.08**	0.65	27.35

**Table 4 sensors-23-07421-t004:** Table comparing the results with and without using the smooth algorithm.

	Original Method	Proposed Method
L [m]	14.90	**13.77**
T [s]	61.18	**40.07**
$e_{tr}$ [m${}^{2}$]	1.89	**0.68**
$a_{avg}$ [m/s${}^{2}$]	0.27	**0.25**
$\kappa_{avg}$ [m${}^{-1}$]	0.58	**0.49**
$T_{init}$ [ms]	46.26	**4.27**

## Data Availability

Not applicable.

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
