# Peer review of "Smooth Autonomous Patrolling for a Differential-Drive Mobile Robot in Dynamic Environments"

_sensors, 2023, doi:10.3390/s23177421_

Round 1

Reviewer 1 Report

This paper presents a smooth motion planning method for patrolling tasks. The proposed method consists of two parts: global planner and local planner. The global part is a user-defined path, and the local planner part uses the TWD * algorithm to avoid collisions. The authors also consider orientation alignment and use clothoid curves for path smoothing. After that, the velocity and angular velocity of the robot were calculated through velocity optimization and trajectory tracking.

The planning framework proposed is complete and the experiments are sufficient, but there are still some shortcomings.

  • In the navigation pipeline, orientation alignment and path smoothing are after TWD *. However, neither of them takes obstacles into consideration. These two parts are not completely following the path of the local planner, as shown in the experiment figures. The authors should Consider the risk of collision.
  • The path of the global planner in this paper is given by the user. If a collision occurs, the local planner will be used for replanning. However, the local planner only uses local information rather than the entire map, which may lead to the robot falling into certain traps such as a dead end and unable to find a path to reach the next target point.
  • In the Experiment sections(4 and 5), authors compared different local planners with the proposed, but did not compare other smoothing algorithms with proposed one. However, comparing with other existing smoothing algorithms can discover the advantages and disadvantages of the algorithm.
  • The statistics in comparison table may be too simple. Only path length, execution time and tracking error are included. More data such as average acceleration, average curvature will facilitate better analysis. In addition, it is mentioned in the paper that the proposed algorithm is "computationally very efficient", so the computation time should also be included.
  • In the orientation alignment part, golden ration is used to generate extra points. However, authors did not explain why the golden ratio was used, and is it better than other extra point methods?

Reviewer 2 Report

The study of the performance of mobile robots on dynamic terrains is an open field for research and production. Therefore, I congratulate the authors on their article. Additionally, the article covers both simulation and experimental tests, which adds value to the paper. Furthermore, to enhance the paper's utility for fellow researchers in reproducing the simulation and production, I suggest considering the following points:

1.Describe the dynamic obstacle behavior and how it changes position during the robot's motion (section 5).

2. Since the proposed method is for industrial applications, adding a section about the limitations of the method and algorithm in real-life implementation is necessary.

3. The article considers a real-time algorithm, not an on-line one. It is better to change all occurrences of "on-line algorithm" to "real-time algorithm" in the abstract, conclusion, and experiment sections (not in the literature review).

The quality of the English is very good, I could not find linguistic issues in the article.

Reviewer 3 Report

The presented article concerns the development of a method for planning a collision-free trajectory of a mobile robot. The proposed Authors' approach combines global and local planning suitable for changing large environments and enabling efficient path replanning with arbitrary robot orientation. Although the article's content is very interesting from the scientific point of view, the manuscript needs a solid conceptual framework based on the literature. Hence, it would be highly appreciated if the Authors could include a separate section on the theoretical framework that serves as the foundation for the study. Additionally, Authors should enhance the section “Conclusion“ by describing the study's limitations. I recommend this research paper for publication after these corrections.

Round 2

Reviewer 1 Report

I thank the authors for addressing all the issues that I raised and providing an exhaustive response. I believe that the paper has been considerably improved as a result of this effort. As such, I do not have further comments on the manuscript.